# A Comprehensive Study of Privacy Risks in Curriculum Learning

## Abstract

Curriculum learning (CL) is a machine learning technique that progressively trains a model on data of increasing difficulty or complexity. This way, the model can learn more efficiently and achieve better performance than random or uniform sampling of data. However, most existing works focus on improving the performance of CL and its privacy risks have never been studied. In this work, we take the first step to investigate the privacy leakage of CL through the lens of membership inference attack (MIA) and attribute inference attack (AIA). Our evaluation of 9 benchmark datasets using various attack methods (NN-based, metric-based, label-only MIA, and NN-based AIA) highlights new insights. First, MIA is slightly more effective with CL, especially on a subset of challenging training samples. Second, models trained with CL are less susceptible to AIA compared to MIA. Third, established defense techniques like DP-SGD, MemGuard, and MixupMMD remain effective under CL, albeit with a notable accuracy impact for DP-SGD. Lastly, we propose a novel MIA called Diff-Cali, which leverages difficulty scores to enhance calibration and effectiveness against all CL and normal training methods. With this study, we hope to draw the community's attention to the unintended privacy risks of emerging machine-learning techniques and develop new attack benchmarks and defense solutions.

## 1 Introduction

The success of machine learning (ML), especially deep learning (DL), hinges on advancements in algorithms, software, and hardware for training models on large-scale datasets. Traditionally, a neural network (NN) is trained by feeding random mini-batches from the training dataset, forcing the NN to "remember" samples in a random order. In contrast, humans learn easy concepts first, guided by curricula. Inspired by the human brain (Rumelhart et al., 1986), curriculum learning (CL) simulates human learning by ordering training data with difficulty scores, repeating this order across training epochs (Bengio et al., 2009). Using a "teacher" network, difficulty scores are assigned to samples, guiding the training process. Previous studies demonstrate CL achieving fast learning speed and high test accuracy (Soviany et al., 2021; Wang et al., 2021), adopted in various domains like computer vision (Bengio et al., 2009; Sakaridis et al., 2019; Duan et al., 2020; Soviany et al., 2020), natural language processing (Bengio et al., 2009; Spitkovsky et al., 2009; Zhou et al., 2020; Guo et al., 2020; Liu et al., 2020), where it claims prominent success (Wang et al., 2021).

ML's success has raised growing concerns about privacy, notably due to sensitive information in training data. The most representative privacy threats are membership inference attacks (MIAs) and attribute inference attacks (AIA). MIA determines if a data point was part of the training set, while AIA infers its sensitive attribute. Recent attacks have highlighted real privacy risks, e.g., over 80% MIA accuracy against CIFAR100 (Salem et al., 2019). Recent studies also reveal varying vulnerability of data samples to these attacks (Yaghini et al., 2019), influenced by target classes (Jia et al., 2019), individuals (Long et al., 2018), and subgroups (Chang & Shokri, 2021). However, existing research assumes standard stochastic training in the target model. Hence, it's crucial to investigate how training techniques affect privacy for the overall population and individual samples. Furthermore, Shumailov et al. (2021) studied the connection between data ordering and backdoor attacks, which indicates data ordering could have negative impacts. This further motivates us to investigate the privacy risks of CL.

**Our Study.** We quantitatively measure the privacy risks of CL. We select two popular CL methods, bootstrapping (Hacohen & Weinshall, 2019) and transfer learning (Weinshall et al., 2018), as the evaluation objects, and construct two other curriculum, named baseline curriculum and anti-curriculum, to understand the impact of data ordering and repeating, respectively. We select 9 real-world, large-scale datasets (6 are image datasets and 3 are tabular datasets), train target models with those CL methods and a normal method, and attack the models with representative MIA and AIA methods.

Regarding MIA, our evaluation shows that the target models become slightly more vulnerable under CL, e.g., the average attack accuracy (trained on ResNet-18 with transfer leaning) on our selected image datasets increase from 0.01% to 2.46%. More importantly, we find CL has a much bigger impact on the samples within the difficult group compared to the easy group, with the biggest gap of 4.23% in terms of attack accuracy for ResNet-18 trained on CIFAR100. We reveal the reason is that the data order reinforces the learning process, hence guiding the model to memorize difficult samples better, which is supported by measuring the memorization scores. Regarding AIA, we find CL does not increase the attack accuracy, which can be explained by the fact that the sensitive attribute to be inferred is not influenced by data ordering and repeating.

Furthmore, we also study existing defenses under the CL settings, including DP-SGD (Abadi et al., 2016), MemGuard (Jia et al., 2019), MixupMMD (Li et al., 2021) and AdvReg (Nasr et al., 2018). The result shows that none can mitigate MIA without dampening the model's accuracy. In particular, DP-SGD is effective in curbing MIA, and the drop in attack accuracy is even more than the normal setting. However, the privacy provided by DP-SGD is at the cost of dropping the classification accuracy of the target model. Inspired by CL and a recent MIA that calibrates membership scores to achieve better attack accuracy (Watson et al., 2022), we consider the difficulty score as input for calibration and propose a new MIA method, named Diff-Cali (difficulty calibrated MIA). Our attack cannot only bring the difficult samples to a more vulnerable stage but also achieve a higher true-positive rate at low false-positive rate regions.

**Contributions.** The contributions of this work are summarized below.

- We take the first step to understanding the privacy risks introduced by CL.
- We conduct a comprehensive analysis to quantify the privacy risks and our results show CL introduces disparate impacts to samples separated by difficulty levels.
- We propose a new MIA that exploits the difficulty scores for better attack performance.

## 2 PRELIMINARY

### 2.1 CURRICULUM LEARNING

Curriculum Learning (CL) emulates human learning by structuring training data, allowing ML models to start with easier samples before progressing to harder ones (Bengio et al., 2009). This accelerates model convergence and boosts testing accuracy (Bengio et al., 2009; Weinshall et al., 2018; Graves et al., 2017; Hacohen & Weinshall, 2019). Weinshall et al. (2018) demonstrated a 0.5% to 3.5% accuracy improvement using transfer learning to construct the curriculum. CL has garnered substantial interest in the ML community and finds applications across various domains. Let $\mathcal{X} = \{X_i\}_{i=1}^N = \{(x_i, y_i)\}_{i=1}^N$ be the training dataset, where $N$ is the number of samples, $x_i$ is a data point, and $y_i$ is the label of $x_i$. $T$ is the ML model to be trained. The standard training procedure will sample $\mathcal{X}$ uniformly to generate the mini-batches. Instead, CL uses a *difficulty measurer* $f(\mathcal{X}, C)$ to generate difficult scores for $\mathcal{X}$, and a *training scheduler* sorts $\mathcal{X}$ by the difficult scores in an ascending order ahead of training. $C$ is the curriculum, and we will elaborate on its common options in Section 4.1. A sequence of subsets $\mathcal{X}'_1, \ldots, \mathcal{X}'_M \subseteq \mathcal{X}$ are extracted from $\mathcal{X}$ after sorting, and the size of $\mathcal{X}'_i$ is determined by a *pacing function* $g(i)$. A mini-batch $\mathcal{B}_i$ is sampled uniformly from $\mathcal{X}'_i$. See algorithm in Appendix E.

### 2.2 PRIVACY RISKS IN MACHINE LEARNING

Previous studies have demonstrated that ML models can inadvertently retain sensitive information from the training data, making them susceptible to attacks like MIA (Shokri et al., 2017; Nasr et al.,

2018; 2019; Salem et al., 2019) and AIA (Melis et al., 2019; Song & Shmatikov, 2020). These attacks have been extensively investigated, and below we give a brief overview.

**Membership Inference Attack (MIA).** Given a target model $T$ and any adversary's external knowledge $K$, the goal of MIA is to determine whether a data sample $x$ was used to train the model. Formally, we have:

$$\mathcal{A}_{MI} : x, T, K \mapsto 1 \text{ or } 0 \tag{1}$$

where $T$ is the target model and $K$ is the adversary's external knowledge, e.g., the distribution of the training data for $T$. 1 (0) denotes the sample is a member (non-member).

MIA can pose significant privacy risks. For instance, in a medical context where a model is trained on clinical records to determine medicine dosage (Jia et al., 2019), MIA could reveal if a person has cancer. Following previous works (Shokri et al., 2017; Salem et al., 2019; Song & Mittal, 2021; Li & Zhang, 2021; Choo et al., 2021), we assume the adversary only has black-box access to $T$. Further details on MIA are outlined in Section 4.2.

**Attribute Inference Attack (AIA).** Unlike MIA, AIA's goal is to infer extra attributes of a data sample, e.g., inferring political views from a model trained for gender classification. These attributes are typically unaddressed during the target model's training. However, due to ML's inherent over-learning property (Song & Shmatikov, 2020), the target model might inadvertently capture irrelevant attributes.

Instead of having direct access to the sample, we follow previous work (Melis et al., 2019; Song & Shmatikov, 2020) and consider the adversary only has its *representation* (e.g., embedding) generated by a target model $T$. Formally, AIA can be defined as:

$$\mathcal{A}_{AI} : h \mapsto s \tag{2}$$

where $h$ is a sample's representation provided by $T$ and $s$ is the sample's sensitive attribute predicted by $\mathcal{A}_{AI}$. Section 4.4 elaborates the details.

## 3 DATASETS AND TARGET MODELS

For evaluation, we choose 9 unique datasets—8 for MIA and 3 for AIA. Among these, six are image datasets, and the remaining three contain non-image data. The target model architectures for the image datasets are ResNet-18, ResNet-34, and MobileNet (He et al., 2016; Sandler et al., 2018). Please refer to Appendix A for details on the datasets, models, and model training settings.

## 4 METHODOLOGY

In this section, we describe the curriculum designs experimented with by our study, the implementation of the basic MIA and AIA, our proposed MIA, and the defense techniques to be tested.

### 4.1 CURRICULUM DESIGNS

We select two popular, openly available CL methods[1][2] for training the target model. Our key findings, outlined in Section 5, are expected to extend to other CL methods like self-paced curriculum (Kumar et al., 2010; Jiang et al., 2015) and automated curriculum (Graves et al., 2017), as they share similar high-level concepts. Self-paced curriculum, for instance, deviates from bootstrapping only by not fully directing the learning process via the curriculum. Below, we elaborate on the two chosen CL methods.

**Bootstrapping (Hacohen & Weinshall, 2019).** The target model $T$ is first trained without CL, then it serves as a difficulty measurer ($f$ in Algorithm 1) to order the training samples by their loss.

**Transfer learning (Weinshall et al., 2018).** Unlike bootstrapping, we employ a pre-trained model, specifically Inception-v3 (Szegedy et al., 2016), to measure difficulty. Inception-v3 is a well-established image recognition model achieving over 78.1% accuracy on ImageNet (Deng et al.,

---

[1] https://github.com/GuyHacohen/curriculum_learning
[2] https://github.com/rsundar96/curriculum-learning-acceleration

2009). However, we did not conduct transfer learning evaluation on tabular datasets due to the absence of a widely recognized pre-trained model in that context.

To thoroughly evaluate the effectiveness and vulnerabilities of the aforementioned CL methods, we introduce two additional comparison methods.

**Baseline curriculum.** This employs a fixed, unrelated curriculum for all training epochs, differing from normal training where a new random order is generated per epoch.

**Anti-curriculum.** It utilizes the bootstrapping difficulty measurer but organizes samples from difficult to easy, reversing bootstrapping's order.

We opt for varied exponential pacing (Hacohen & Weinshall, 2019), incrementing the data fraction exponentially at each step, as suggested by Hacohen & Weinshall (2019). Various pacing functions exhibit comparable performance. Table 3 (see Appendix A) validates the effectiveness of CL, with at least one CL method consistently outperforming normal training by 0.06% to 4.42%.

As described in Section 2.1, CL can accelerate the training process to reach higher accuracy. We verify this in Appendix B.

Additionally, CL is expected to affect classification accuracy differently across various samples. Apart from the analysis in Section 5, we employ t-distributed stochastic neighbor embedding (t-SNE) for visualization. Further details, including the visualization, can be found in Appendix F.3.

## 4.2 BASIC MIA

For the existing MIAs, we consider three well-known attacks: NN-based (Neural Network-based) (Shokri et al., 2017; Salem et al., 2020), metric-based (Song & Mittal, 2021), and label-only attacks (Li & Zhang, 2021; Choo et al., 2021). See Appendix C for details.

**MIA Models.** Following the original setting of the NN-based attacks (Shokri et al., 2017), we adopt a 3-layer MLP with 64 and 32 hidden neurons, and 2 output neurons, as our attack model $\mathcal{A}_{MI}$. We use cross-entropy as the loss function and Adam as the optimizer with a learning rate of 0.01. $\mathcal{A}_{MI}$ is trained for 100 epochs. For metric-based attacks, we follow the implementation of Song & Mittal (2021) and consider 4 metrics, including correctness, confidence, entropy, and modified entropy. The associated attack methods are named metric-corr, metric-conf, metric-ent, and metric-ment. For label-only attacks, we leverage the implementation from ART (Nicolae et al., 2018).

Related research has shown that NN-based attacks often, though not universally, achieve better attack accuracy compared to metric-based and label-only attacks (Shokri et al., 2017; Salem et al., 2019; He & Zhang, 2021). Thus we use NN-based attack (specifically black-box-top3) for most of our evaluation in Section 5.

## 4.3 OUR PROPOSED MIA

Since CL orders training samples by the difficulty levels, which affects the trained model, we are interested in whether MIA can be enhanced when the target model is trained under CL. To this end, we propose a *new* MIA method (termed Diff-Cali) that is customized against CL. Below we first introduce calibrated MIA that inspires the design of Diff-Cali, and then the details of Diff-Cali.

**Calibrated MIA.** Recently, Watson et al. (2022) proposed to use a calibrated membership score instead of the standard membership score (e.g., loss) to determine whether a sample is a member. Assume $s(T, x)$ is the original membership score, where $T$ is the target model and $x$ is a sample. The calibrated membership score $s_{cal}(T, x)$ is defined as follows:

$$s_{cal}(T, x) = s(T, x) - \mathbb{E}_{\mathcal{S} \leftarrow \mathcal{A}(\mathcal{D})}[s(\mathcal{S}, x)] \tag{3}$$

where $\mathcal{S}$ are shadow models[3] that behave similarly as $T$, $\mathcal{D}$ is the shadow dataset, function $s(T, x)$ and $s(\mathcal{S}, x)$ output the membership scores from target and shadow models, $\mathcal{A}$ randomly samples

---

[3]$\mathcal{S}$ are named as reference models in Watson et al. (2022), which resemble shadow models (Shokri et al., 2017) as they are also trained on the same data distribution of $T$.

subsets of $\mathcal{D}$ to train $\mathcal{S}$, and $\mathbb{E}$ computes the expectation of $s(\mathcal{S}, x)$. Finally, $s_{cal}(T, x)$ is compared to a fixed threshold $\theta$ and a sample is considered a member if $s_{cal}(T, x) \geq \theta$.

Previous MIA methods could have high false positive rate (FPR) on non-members, which are often over-represented in the samples to be tested by the attacker. Equation 3 addresses this issue by using the *difference* between the target model and shadow models to derive the membership signal: if $x$ is non-member to $\mathcal{S}$, it is also more likely non-member to $T$, therefore $s_{cal}(T, x)$ should be small. The evaluation results in Watson et al. (2022) shows the area under ROC curve (AUC) can be improved "by up to 0.10" (e.g., after calibrating the loss-based membership score with Equation 3).

**Difficulty Calibrated MIA (Diff-Cali).** Calibrated MIA compares $s_{cal}(T, x)$ of all samples to a fixed threshold $\theta$, and we argue that $\theta$ can be *calibrated as well*. We observe that a CL curriculum re-orders the samples by their difficulty before the target model is trained and such strategy changes how a sample is memorized and vulnerable under MIA (see Section 5.1 and Section 5.2). More specifically, we observe that CL makes the target model more vulnerable to MIA, especially for difficult samples (Finding 1 in Section 5.1). Therefore, we can update $\theta$ according to the curriculum and make the attack model more accurate. We assume the attacker can generate a curriculum similar as the one used by the target model. For example, the attacker can use the publicly released pre-trained model to generate the curriculum. Alternatively, the attacker can train shadow models that are similar as the target model, then builds curriculum according to loss from them.

We implement this idea for NN-based MIA. When the attack model $\mathcal{A}_{MI}$ outputs the prediction posteriors for an input $x$, the posterior of the label "member" is compared against $\theta$, and $x$ is predicted as member when the posterior is larger. When training $\mathcal{A}_{MI}$, we adjust $\theta$ based on samples' difficulty level to improve the training accuracy, and the the pseudo-code is shown in Appendix E. Specifically, in each epoch, the calibrated membership scores $s_{cal}(T, \mathcal{D})$ are generated for $\forall x \in \mathcal{D}$, and we use the loss to compute $s$. Next, we try to find the threshold $\theta_0$ (ranging from 0 to 0.1 based on our empirical study) that achieves the best accuracy in separating members and non-members from $\mathcal{D}$. After that, $\mathcal{A}_{MI}$ is updated by minimizing the training loss on $\mathcal{D}$ through adjusting the threshold with the following function:

$$g(x, C, \theta_0) = \frac{(|\mathcal{D}| - C(x))(\theta_0 - 0.0001)}{|\mathcal{D}| - 1} + 0.0001 \tag{4}$$

where $C(x)$ indicates the rank of sample $x$ given by curriculum $C$. The rank for the easiest sample is 1 while the most difficult is $|\mathcal{D}|$. $g(x, C, \theta_0)$ is to assign a threshold $\theta$ from $[0.0001, \theta_0]$ (0.0001 is the initial threshold suggested by Watson et al. (2022)) to each $x$ based on its difficulty level (determined by a curriculum $C$), that is, calibrating threshold of each $x$ based their difficulty level. The most difficult sample compares to 0.0001, the easiest one compares to $\theta_0$, and others compare to $\theta$ that is ranged in $[0.0001, \theta_0]$. The more difficult $x$ have smaller threshold, meaning that we are lowering the bar for them to be predicted as members comparing to the easy samples. During the testing phase, the threshold for a sample $x$ is also adjusted with $g(x, C, \theta_0)$.

## 4.4 BASIC AIA

Song et al. proposed an inference-time attack and model-repurposing attack (Song & Shmatikov, 2020) for AIA, and here we focus on the first attack and follow the same setting as this work. We consider the model evaluation to be partitioned (Song & Shmatikov, 2020) or the model is trained under federated learning (Melis et al., 2019). The target model $T$ is split into two parts, i.e., an encoder and a classifier, and the adversary has black box access to the encoder $E$. The attacker has an auxiliary dataset $D$ containing pairs of $(x, s)$ where $s$ is the sensitive attribute. The embeddings $h$ can be generated by querying $E$, i.e., $h = E(x), \forall x \in D$. All pairs of $(h, s)$ will be used to train the attack model $\mathcal{A}_{AI}$ and later used to predict the values of $s$ in the target model $T$.

**AIA Model.** Our $\mathcal{A}_{AI}$ is a 3-layer MLP with 128 hidden neurons in each hidden layer. We use cross-entropy as the loss function and SGD as the optimizer with a learning rate of $0.01$. The attack model is trained for 100 epochs. The dimension of each sample's embedding (i.e., the second to the last layer's output) is 512 for ResNet-18, 512 for ResNet-34, and 1024 for MobileNet. To train the target model $T$, we use the label for the original classification task (e.g., gender). To train $\mathcal{A}_{AI}$, we use the label from another field (e.g., race).

| Method
Dataset | Normal | Bootstrapping | Anti-curriculum | Baseline | Transfer Learning |
|---|---|---|---|---|---|
| Tiny ImageNet | $0.9193 \pm 0.0000$ | $0.9385 \pm 0.0000$ | $0.9116 \pm 0.0001$ | $0.9207 \pm 0.0000$ | $\mathbf{0.9439} \pm 0.0000$ |
| CIFAR100 | $0.8577 \pm 0.0011$ | $\mathbf{0.8751} \pm 0.0001$ | $0.8376 \pm 0.0001$ | $0.8582 \pm 0.0001$ | $0.8718 \pm 0.0001$ |
| Place100 | $0.9425 \pm 0.0000$ | $0.9549 \pm 0.0001$ | $0.9335 \pm 0.0001$ | $0.9416 \pm 0.0001$ | $\mathbf{0.9617} \pm 0.0001$ |
| Place60 | $0.8773 \pm 0.0022$ | $\mathbf{0.8987} \pm 0.0001$ | $0.8625 \pm 0.0001$ | $0.8827 \pm 0.0001$ | $0.8902 \pm 0.0001$ |
| SVHN | $0.5570 \pm 0.0000$ | $\mathbf{0.5605} \pm 0.0002$ | $0.5514 \pm 0.0001$ | $0.5599 \pm 0.0003$ | $0.5580 \pm 0.0003$ |
| Purchase | $\mathbf{0.9524} \pm 0.0016$ | $0.9453 \pm 0.0024$ | $0.9118 \pm 0.0122$ | $0.9458 \pm 0.0015$ | - |
| Texas | $0.6749 \pm 0.0092$ | $\mathbf{0.7068} \pm 0.0139$ | $0.5950 \pm 0.0161$ | $0.7039 \pm 0.0122$ | - |
| Location | $0.9153 \pm 0.0066$ | $\mathbf{0.9194} \pm 0.0048$ | $0.8980 \pm 0.0038$ | $0.9169 \pm 0.0038$ | - |

Table 1: Accuracy of NN-based MIA on models trained on 8 datasets. Transfer learning CL does not apply to non-image dataset Purchase, Texas and Location.

| Method
Attack | Normal | Bootstrapping | Anti-curriculum | Baseline | Transfer Learning |
|---|---|---|---|---|---|
| NN-based (Shokri et al., 2017) | $0.8572 \pm 0.0011$ | $\mathbf{0.8751} \pm 0.0001$ | $0.8376 \pm 0.0002$ | $0.8582 \pm 0.0001$ | $0.8718 \pm 0.0001$ |
| Metric-corr (Song & Mittal, 2021) | $0.6920 \pm 0.0000$ | $0.6820 \pm 0.0000$ | $0.6905 \pm 0.0000$ | $\mathbf{0.6930} \pm 0.0000$ | $0.6855 \pm 0.0000$ |
| Metric-conf (Song & Mittal, 2021) | $0.8600 \pm 0.0000$ | $\mathbf{0.8810} \pm 0.0000$ | $0.8458 \pm 0.0000$ | $0.8553 \pm 0.0000$ | $0.8740 \pm 0.0000$ |
| Metric-ent (Song & Mittal, 2021) | $0.8490 \pm 0.0000$ | $\mathbf{0.8750} \pm 0.0000$ | $0.8320 \pm 0.0000$ | $0.8435 \pm 0.0000$ | $0.8685 \pm 0.0000$ |
| Metric-ment (Song & Mittal, 2021) | $0.8620 \pm 0.0000$ | $\mathbf{0.8820} \pm 0.0000$ | $0.8463 \pm 0.0000$ | $0.8568 \pm 0.0000$ | $0.8760 \pm 0.0000$ |
| Label-only (Nicolae et al., 2018) | $0.8200 \pm 0.0082$ | $\mathbf{0.8263} \pm 0.0082$ | $0.7963 \pm 0.0117$ | $0.8050 \pm 0.0045$ | $0.8088 \pm 0.0074$ |
| Cali (Watson et al., 2022) | $0.7889 \pm 0.0012$ | $\mathbf{0.8272} \pm 0.0009$ | $0.7532 \pm 0.0004$ | $0.7781 \pm 0.0025$ | $0.8148 \pm 0.0013$ |
| Diff-Cali | $0.8519 \pm 0.0003$ | $\mathbf{0.8670} \pm 0.0006$ | $0.8382 \pm 0.0006$ | $0.8438 \pm 0.0008$ | $0.8614 \pm 0.0006$ |

Table 2: Average accuracy of NN-based, metric-based, label-only and our Diff-Cali attacks on models trained on CIFAR100 with ResNet-18.

## 4.5 DEFENSE METHODS

Some defense methods have been proposed to reduce the success rate of privacy attacks, in particular, MIA. We are interested in how they perform under curriculum learning and our proposed attack and we select DP-SGD (Abadi et al., 2016), MemGuard (Jia et al., 2019), MixupMMD (Li et al., 2021) and AdvReg (Nasr et al., 2018). DP-SGD and MemGuard represent two directions in privacy protection, while MixupMMD and AdvReg are two more recent defense methods. See Appendix D for details.

## 5 EVALUATION RESULTS

**Evaluation setup.** For evaluating MIA and AIA, we partition each dataset as outlined in Section 3: one part for target model training, one for shadow model training, and one for testing both models. For defense methods, we split each dataset into five parts, accommodating reference datasets for certain advanced methods. Additional defense method details can be found in Appendix F.6.

**Evaluation metrics.** We use accuracy to evaluate the MIA and AIA, as well as the impact of curriculum learning and defenses. Besides, following Carlini et al. (2021), we compute the true-positive rate (TPR) at the false-positive rate (FPR) of the attacks.

## 5.1 EVALUATION OF BASIC MIA

We start with the experiments on the 5 image datasets using ResNet-18 as the target model architecture and evaluate tabular datasets. See Appendix F for more evaluations, i.e., loss distribution, different model architectures, non-image datasets, and defenses.

**MIA Accuracy.** We first find models trained with meaningful CL methods like bootstrapping and transfer learning are slightly more vulnerable to MIA. Table 1 presents the largest improvement in attack accuracy (NN-based) for image datasets is 2.46% (TinyImageNet with transfer learning), and for non-image datasets, it's 3.20% (Texas with bootstrapping). Bootstrapping and transfer learning show the most vulnerability, with an average of 1.29% and 1.44% improvement in attack accuracy against normal training, respectively. For baseline CL, attack accuracy decreases for Place100, while a slight increase is observed for other datasets. In anti-curriculum CL, attack accuracy decreases for all datasets. This suggests that both data repeating (as seen in baseline results) and ordering (as seen in bootstrapping and anti-curriculum results) in CL contribute to vulnerability to MIA (explained

in Section 4.1). The consistent performance of bootstrapping and anti-curriculum highlights **the significant role of data ordering**.

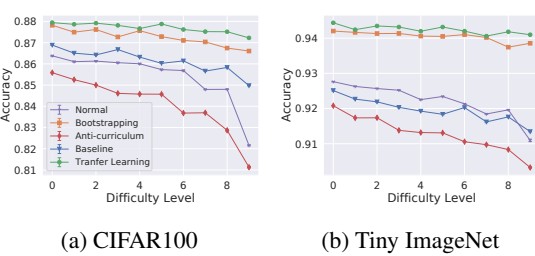

(a) CIFAR100  (b) Tiny ImageNet

Figure 1: MIA accuracy on CIFAR-100, Tiny ImageNet. ResNet-18 is used for target model training.

For metric-based and label-only attacks, the results align with the NN-based attack, as shown in Table 2. The only exception is metric-corr performing worse compared to other attacks with bootstrapping. This outcome can be attributed to metric-corr's assumption that the target model is trained to predict correctly on its training data, which might not generalize well to the test data.

Figure 1 shows the attack accuracy of samples from different difficulty levels. More specifically, we construct the test dataset as half member samples and half non-member samples. Member samples are divided into different difficulty levels while non-member samples across each difficulty level are fixed. Figure 1 demonstrates that using meaningful curriculum (i.e., bootstrapping and transfer learning) makes the model more vulnerable, especially for the difficult samples.

**Confidence Score.** Since the key contribution of CL is to factor in the samples' difficulty levels

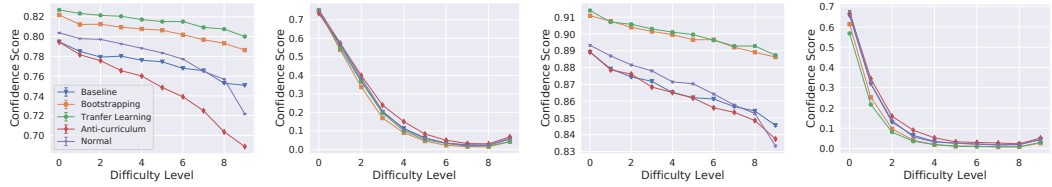

(a) Members of CIFAR100 (b) Non-members of CIFAR100 (c) Members of Tiny ImageNet (d) Non-members of Tiny ImageNet

Figure 2: Attack model's confidence score for both member and non-member samples on CIFAR-100 and Tiny ImageNet. ResNet-18 is used for target model training, and data samples are arranged according to their difficulty scores from bootstrapping.

during the training, we evaluate how difficulty levels impact the samples' vulnerability individually.

Figure 2 depicts the attack model's confidence score by samples' difficulty levels. Notably, difficult samples aren't notably more vulnerable than easy ones, but the gap in confidence scores is narrower, particularly for member samples. For instance, with the CIFAR100 target model, the attack model identifies the most difficult member samples (level 9 difficulty) with over 7.83% higher confidence due to transfer learning (Figure 2a). Interestingly, anti-curriculum can even yield a higher confidence score for the most difficult member samples compared to normal training (Figure 2c). This suggests that presenting difficult samples early in training doesn't necessarily increase the likelihood of the model forgetting them.

**TPR at Low FPR.** We here measure the relation between TPR at low FPR. We present the ROC curve for the attacks with both linear scaling and log scaling to emphasize the low-FPR regime. Figure 3a and Figure 3b demonstrate the ROC curve for NN-based attack. The results show that using curriculum increases ROC. The TPR of transfer learning and bootstrapping are generally higher than the others except at extremely low FPR ($< 10^{-4}$). This indicates CL introduces disparate impact to members and non-members for most samples.

## 5.2 ANALYSIS WITH DATA MEMORIZATION

The previous experiments show CL makes the difficult samples more vulnerable. Here, we attempt to explain this observation with a more principled analysis. Recent works (Feldman, 2020; Feldman & Zhang, 2020) suggest the effectiveness of MIA could be tied to how well the target model

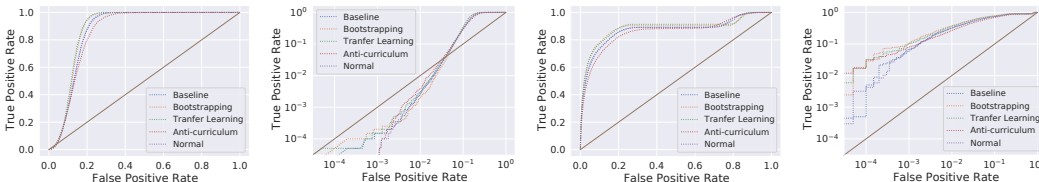

(a) Linear scale NN-based    (b) Log scale NN-based    (c) Linear scale Diff-Cali    (d) Log scale Diff-Cali

Figure 3: TPR/FPR of NN´based MIA and Diff-Cali under different training method trained with ResNet-18 on CIFAR100.

*memorizes* individual data sample. The notion of memorization is formally defined as Feldman (2020):

$$\mathbf{mem}(\mathcal{A}, \mathcal{D}, i) := \Pr_{T \sim A(D)} \left[ T\left(x_i\right) = y_i \right] - \Pr_{T \sim A\left(D^{\setminus i}\right)} \left[ T\left(x_i\right) = y_i \right] \tag{5}$$

where $\mathcal{A}$ denotes the training algorithm, $\mathcal{D}$ denotes the training dataset, $T$ is the trained model, $(x_i, y_i)$ denotes one sample with its ground-truth label, and $\mathcal{D}^{\setminus i}$ denotes $\mathcal{D}$ with $i$-th sample removed. The model is likely to memorize the data sample if adding $(x_i, y_i)$ to training significantly changes the model's prediction on $y_i$. Though Equation 5 models the memorization of a single data sample, we can easily extend it to quantify the memorization of multiple samples at once.

We evaluate ResNet-18 trained with CI-FAR100. First, we exclude 800 most diffcult samples and train a model without these data via bootstrapping (labeled as "not seen"). Then, we train the model under CL, focusing on data memorization, with the 800 data samples placed either at the start ("first seen"), end ("last seen"), or at random positions ("random") in each training epoch. Figure 4 depicts the prediction probability of the true labels of the 4 scenarios. To assess data memorization, we compare "first seen," "last seen," and "random" with "not seen" using the concept from Equation 5. Notably, apart from "not seen," the other three scenar-

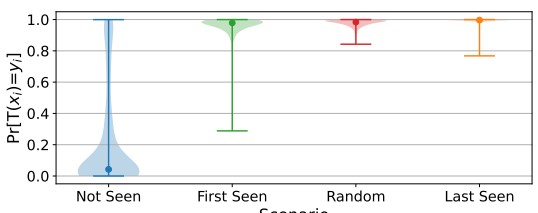

Figure 4: Memorization: violin plots of prediction probability of 800 most difficult samples, according to bootstrapping CL.

ios exhibit strong memorization of difficult samples, evident from higher prediction probabilities for the true class. Data ordering significantly influences data memorization; notably, "last seen" demonstrates the strongest memorization compared to "first seen" and "random." The vulnerability of difficult samples under CL is attributed to enhanced memorization facilitated by specific data ordering.

## 5.3 EVALUATION OF DIFF-CALI

In order to fully utilize the information of difficulty levels exposed by CL, we propose Diff-Cali as described in Section 4.3. Overall, NN-based attack still has a slightly better attack accuracy compared to Diff-Cali, but Diff-Cali has higher confidence scores for difficult samples and has better TPR at low FPR regime.

**Attack Accuracy.** Table 2 presents the accuracy of Diff-Cali, which is about 1% lower compared to NN-based attack on all CL methods. Figure 5 depicts the attack accuracy on CIFAR100 and Tiny ImageNet. ThoughDiff-Cali achieves slightly lower (less than 1.44%) accuracy compared to NN-based attack, with adaptive calibration, we are able to make **the difficult samples more vulnerable**: e.g., the attack accuracy of difficulty level at 9 and 0 are 86.47% and 86.32% for transfer learning under CIFAR100. The most difficult samples now can be predicted 2.64% and 2.350% more accurately for normal and anti-curriculum ML, respectively. Overall, Diff-Cali is able to overcome the privacy risk discrepancy of different samples through calibration and results in better attack accuracy for difficult samples for normal ML and anti-curriculum ML.

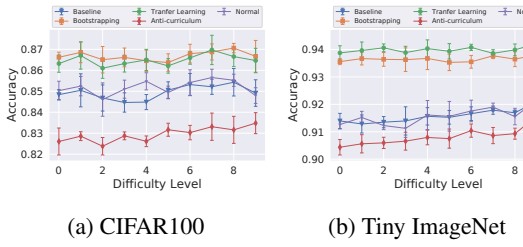

(a) CIFAR100      (b) Tiny ImageNet

Figure 5: Diff-Cali's accuracy for models trained on CIFAR100 and Tiny ImageNet with ResNet-18.

**Confidence Score.** In our MIA evaluation (see Figure 13 in Appendix F), member samples consistently achieve high confidence scores (CIFAR100: $>0.7807$, Tiny ImageNet: $>0.8678$), outperforming NN-based approaches for Tiny ImageNet (Figure 2). In summary, we improve member confidence by 3.29% for CIFAR100 and 3.45% for Tiny ImageNet, while reducing non-member confidence (lower scores imply lower misclassification risk) by 0.0414 for CIFAR100 and 0.1751 for Tiny ImageNet. Unlike prior NN-based attacks, Diff-Cali's accuracy doesn't directly align with confidence scores, emphasizing a unique membership status prediction strategy.

**TPR at Low FPR.** In Figure 3, we show that Diff-Cali can achieve much higher TPR at low FPR ($< 10^{-4}$). We present the ROC curve for the attacks with both linear scaling and log scaling to emphasize the low-FPR regime. Figure 3c and Figure 3d demonstrate the ROC curve for Diff-Cali. The results show that using curriculum increases ROC (Figure 3a, Figure 3c). We observe that our proposed Diff-Cali performs better at low FPR. More specifically, Figure 3b shows that NN-based attack fails to achieve a TPR better than random chance at any FPR below $0.045$ while Diff-Cali can be better than random guessing at all times.

### 5.4 EVALUATION OF AIA

We evaluate four CL methods and normal training under the AIA attack. Table Figure 6 demonstrates that CL does not heighten the vulnerability of the target model. This contrasts with a recent study (He & Zhang, 2021), indicating increased vulnerability under AIA with contrastive learning in specific training settings. Interestingly, normal training yields the highest average attack accuracy (e.g., 0.107 for Place100), even surpassing anti-curriculum. Notably, UTKFace exhibits higher attack accuracy due to its already elevated baseline accuracy (42.1%).

| Dataset / Method | Place100 | Place60 | UTKFace |
|---|---|---|---|
| Normal | **0.107**±0.003 | **0.173**±0.002 | **0.528**±0.005 |
| Bootstrapping | 0.092±0.003 | 0.168±0.004 | 0.515±0.006 |
| Transfer Learning | 0.104±0.001 | 0.150±0.005 | 0.512±0.006 |
| Baseline Curriculum | 0.079±0.004 | 0.143±0.001 | 0.506 ±0.008 |
| Anti-Curriculum | 0.033±0.001 | 0.128±0.005 | 0.517±0.007 |

Figure 6: Average accuracy of AIA ($\pm$ standard deviation) on model trained with different methods. ResNet-18 is the target model architecture.

Further investigation reveals consistent attack accuracy across samples with different difficulty levels (see Figure 15 in Appendix F). This suggests that sample attributes are inherently complex and challenging to learn. The difficulty score, such as bootstrapping, relies on the original ML task, emphasizing the specific attribute targeted for classification. Consequently, data ranking's effectiveness is confined to the chosen classification attribute and does not impact the inference of the intended sensitive attribute.

## 6 CONCLUSION

In this work, we perform the first quantitative study to understand how curriculum learning, a widely-used technique that accelerates model training, affects the privacy of the trained model. Specifically, we trained target models under 6 image datasets and 3 tabular datasets, and performed membership inference attacks (MIA) and attribute inference attacks (AIA) against them to assess the privacy risk in curriculum learning. Our results show that the target model becomes slightly more vulnerable to MIA but not so under AIA. We also found MIA has a significantly larger impact on samples with high difficulty levels. By exploiting the leakage from difficulty levels, we design a new MIA, termed Diff-Cali, which achieves similar overall accuracy with much better TPR at low FPR and can infer difficulty samples from normal ML more accurately. Finally, we evaluate the existing defenses DP-SGD, MemGuard, MixupMMD, and AdvReg in the setting of curriculum learning, and our results show that they are still effective against the basic MIA. With this study, we hope to draw attention to the unintended effects of the emerging machine-learning techniques, and more theoretical analysis into the trade-off between privacy, accuracy, and fairness.

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

# A  DATASET AND MODELS DESCRIPTION

## A.1  MIA DATASETS

We use the following 8 datasets, which are also adopted by previous works (He & Zhang, 2021; Liu et al., 2021; Mireshghallah et al., 2020; Shokri et al., 2017) to study MIA. They are CIFAR100 (Krizhevsky et al., 2009), Tiny ImageNet (Le & Yang, 2015), Place100, Place 60 (Zhou et al., 2017), SVHN (Netzer et al., 2011), Purchase(Shokri et al., 2017), Texas hospital stays (Shokri et al., 2017) and Locations (Yang et al., 2016). We focus on image datasets mainly (the first 5 datasets), but tabular datasets are also evaluated.

**CIFAR100 (Krizhevsky et al., 2009).**  This dataset consists of $60,000$ colored images in $100$ classes, with 600 images per class. The size of each image is $32 \times 32$.

**Tiny ImageNet (Le & Yang, 2015).**  This is a subset of the ImageNet dataset Deng et al. (2009). It contains $100,000$ colored images of 200 classes (500 for each class). The size of each image is $64 \times 64$.

**Place100.**  This dataset is a subset of Places365 (Zhou et al., 2017) dataset, which is composed of more than 1.8 million images with 365 scene categories. Place100 is generated by randomly selecting 100 scene categories with 600 random images per category.

**Place60.**  This dataset is similar to Place100, except that it has 60 classes containing $1,000$ images each.

**SVHN (Netzer et al., 2011).**  The Street View House Numbers (SVHN) dataset is a real-world image dataset containing over $600,000$ digit images. This dataset includes images of house numbers taken from Google Street View images. The size of each image is $32 \times 32$.

**Purchase.**  This is a tabular dataset about purchase styles. Following Shokri et al. (2017), we leverage the Purchase-100 dataset (abbreviated as Purchase) and uses $10,000$ records for training. The dataset itself contains $197,324$ records from 100 classes where each record has 600 binary features.

**Texas hospital stays.**  This dataset contains the information about inpatients stays in several health facilities. Following Shokri et al. (2017), our task is to predict a patient's main procedure. After pre-processing, the resulting dataset has $67,330$ records and 6,170 binary features.

**Locations (Yang et al., 2016) .**  The original dataset was released by Foursquare about its mobile users' location "check-ins", which has 11,592 users and 1,136,481 check-in records. Our task is to predict the user's geo-social type (128 in total). Here we use the version pre-processed by Shokri et al. (2017), which has 446 binary features.

## A.2  AIA DATASETS

Datasets with multiple attributes are required for AIA. To this end, we adapt Place100 and Place60 used as MIA datasets to AIA setting as they both contain multiple attribute labels. More specifically, the model for Place100 outputs whether a sample is an indoor scene, while the sensitive attribute is the category of the scene, which contains 100 labels. Place60 has the total number of categories as 60. In addition to Place100 and Place60, we introduce UTKFace (Zhang et al., 2017) specifically for AIA study.

**UTKFace (Zhang et al., 2017).**  This is a large-scale facial dataset, which consists of over $20,000$ face images with annotations of age, gender, and ethnicity. In our evaluation, we set gender classification as the the task for target model, and the sensitive attribute to be inferred is ethnicity, which includes 5 classes.

## A.3  MODELS

The target model architectures for the image datasets are ResNet-18, ResNet-34, and MobileNet (He et al., 2016; Sandler et al., 2018). We choose these models because variants of ResNet are still achieving SOTA (State of The Art) or near SOTA performance in image classification, and MobileNet is widely used on mobile devices. We adopt cross entropy as the loss function and SGD as

| Method
Dataset | Normal | Bootstrapping | Anti-curriculum | Baseline | Transfer Learning |
|---|---|---|---|---|---|
| Tiny ImageNet | $0.3842 \pm 0.0027$ | $\mathbf{0.4002} \pm 0.0043$ | $0.3776 \pm 0.0036$ | $0.3798 \pm 0.0035$ | $0.3803 \pm 0.0043$ |
| CIFAR100 | $0.6081 \pm 0.0053$ | $\mathbf{0.6232} \pm 0.0078$ | $0.5991 \pm 0.0098$ | $0.6099 \pm 0.0045$ | $0.6127 \pm 0.0221$ |
| Place100 | $0.2992 \pm 0.0054$ | $\mathbf{0.3159} \pm 0.0059$ | $0.2967 \pm 0.0037$ | $0.3088 \pm 0.0060$ | $0.3007 \pm 0.0053$ |
| Place60 | $0.4756 \pm 0.0041$ | $\mathbf{0.4903} \pm 0.0040$ | $0.4815 \pm 0.0025$ | $0.4847 \pm 0.0071$ | $0.4707 \pm 0.0154$ |
| SVHN | $0.9592 \pm 0.0004$ | $0.9598 \pm 0.0006$ | $0.9566 \pm 0.0005$ | $0.9593 \pm 0.0006$ | $\mathbf{0.9599} \pm 0.0006$ |
| Purchase | $0.4931 \pm 0.0055$ | $\mathbf{0.5324} \pm 0.0037$ | $0.4760 \pm 0.0055$ | $0.5289 \pm 0.0043$ | - |
| Texas | $0.4809 \pm 0.0072$ | $\mathbf{0.4975} \pm 0.0066$ | $0.4606 \pm 0.0101$ | $0.4877 \pm 0.0095$ | - |
| Location | $0.5861 \pm 0.0107$ | $\mathbf{0.5914} \pm 0.0027$ | $0.5563 \pm 0.0156$ | $0.5838 \pm 0.0077$ | - |

Table 3: Target model's average test accuracy on different datasets. ResNet-18 is used for all image datasets, and MLP for non-image datasets Purchase, Texas, and Location. Transfer learning CL does not apply to non-image datasets. The target model accuracy is relatively low except for SVHN because we use a subset of the original training data.

the optimizer. We train all models for 200 epochs with a batch size of 128. The learning rate is set to $0.1$[4]. For the non-image dataset Purchase and Location, we choose a 3-layer MLP with the same number of epochs and batch size. The number of neurons in the hidden layer is 256. For Texas dataset, we use 5-layer MLP with 512 neurons in the hidden layer because this dataset contains more features. To avoid fortuitous outcomes, all experiments are repeated 5 times with the standard deviation presented. Table 3 shows the average accuracy of models trained on various datasets.

## B  CL PERFORMANCE

| Dataset
Method | Tiny ImageNet | CIFAR100 | Place100 | Place60 | SVHN | Purchase | Texas | Location |
|---|---|---|---|---|---|---|---|---|
| Normal | 100.0 | 100.0 | 100.0 | 100.0 | 100.0 | 100.0 | 96.770 | 100.0 |
| Bootstrapping | 100.0 | 100.0 | 100.0 | 99.996 | 100.0 | 100.0 | 94.030 | 100.0 |
| Transfer | 100.0 | 99.997 | 100.0 | 99.972 | 100.0 | / | / | / |
| Baseline | 100.0 | 99.993 | 100.0 | 100.0 | 100.0 | 99.990 | 95.600 | 100.0 |
| Anti-curriculum | 99.963 | 100.0 | 100.0 | 99.918 | 100.0 | 100.0 | 97.410 | 100.0 |

Table 4: The average training accuracy of datasets in Table 3. Image datasets are trained on ResNet-18 while non-image datasets are trained on MLP. Numbers are all in percentage. We observe that all training accuracies are nearly 100%. Note that for non-image datasets, we skip the transfer method as there is no a commonly used pre-train model for tabular dataset.

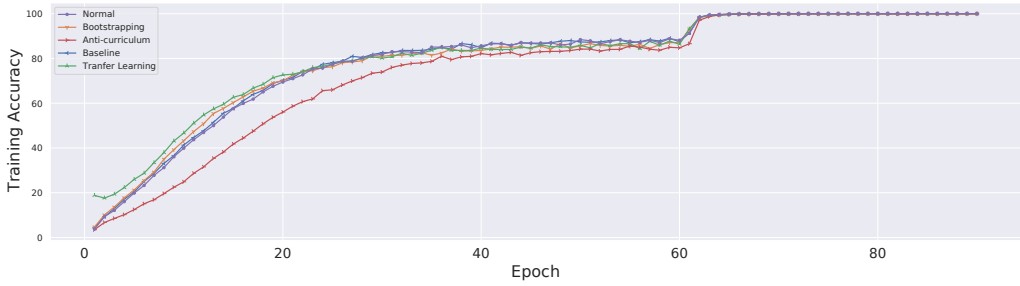

Figure 7: The training accuracy of different training methods with ResNet-18 on CIFAR100 along the increase of epochs (total of 90 epochs).

Average training accuracy details can be found in Table 4. The maximum standard deviation in Figure 3 is 0.0221, with 32 out of 37 results having a standard deviation below 0.01, indicating significant differences among CL methods. Notably, bootstrapping and transfer learning consistently outperform normal training, while anti-curriculum consistently performs the worst. We find it intriguing that the baseline performs equally well as transfer learning curriculum for Place100 and

---

[4]This learning rate is empirically chosen and has a very limited effect on attack accuracy. For example, when using a learning rate of $0.001$, the MIA accuracy is affected by less than $0.2\%$ when attacking a ResNet-18 model trained on CIFAR100.

Place60. This suggests that the transfer learning curriculum may not be the most suitable for these datasets. Figure 7 validates a key motivation for adopting CL: achieving higher accuracy faster. Notably, bootstrapping and transfer learning consistently reach higher accuracy faster than all other methods. Conversely, anti-curriculum takes the longest to achieve the same training accuracy as compared to other methods, underscoring the benefits of a meaningful data order in training. This observation aligns with prior research (Wu et al., 2021; Hacohen & Weinshall, 2019).

## C    BASIC MIAS

NN-based attack assumes a vector of *prediction posteriors* (e.g., confidence scores or loss) of all class labels can be returned by the target model $T$ when querying $T$ with a data sample $x$. It is also assumed that the adversary has a *shadow dataset* ($\mathcal{D}$) that has the same distribution and format as $T$'s private training dataset. $\mathcal{D}$ is used to train a set of *shadow models* $\mathcal{S}$ that behave similarly as $T$ (e.g., having the same architecture as $T$ like previous work (Shokri et al., 2017; Salem et al., 2019; Song & Mittal, 2021)).

The attacker trains an *attack model* $\mathcal{A}_{MI}$ using $\mathcal{S}$. In particular, the attacker queries every shadow model $\mathcal{S}$ with the samples from its own training dataset and a disjoint testing dataset. The prediction posteriors of all samples and whether they are in training (denoted member) or testing (denoted non-member) are used as input to train $\mathcal{A}_{MI}$. Finally, the attacker queries $T$ with a sample of interest $x$ and uses the prediction posteriors as the input to $\mathcal{A}_{MI}$ to predict the membership status.

Compared to the NN-based attack, the model $\mathcal{A}_{MI}$ of metric-based attacks does not need to be trained. Instead, $\mathcal{A}_{MI}$ generates a privacy risk score from the output of $T$ and compares it to class-specific thresholds.

For the label-only attack, it assumes only the prediction label instead of the prediction posteriors are returned from $T$. Still, the adversary can continuously add adversarial perturbations to the input sample $x$ until its prediction label has been changed. The key insight is that the magnitude of the adversarial perturbation is larger for the member sample as $T$ gives a more confident prediction. $\mathcal{D}$ and $\mathcal{S}$ can be used to select a threshold to separate the perturbation magnitudes of members and non-members.

## D    DEFENSE METHODS

**DP-SGD.**  Differentially-Private Stochastic Gradient Descent (DP-SGD) modifies the stochastic gradient descent (SGD) algorithm and integrates $(\epsilon, \delta)$-DP (Dwork et al., 2006) to provide provable privacy guarantee.

**Definition 1.** *($(\epsilon, \delta)$-DP) An algorithm $\mathcal{M}(\cdot)$ satisfies $(\epsilon, \delta)$-differential privacy ($(\epsilon, \delta)$-DP), if and only if for any pair of datasets $V$ and $V'$ that differs in only one element and for any possible output set $O$*

$$\Pr\left[\mathcal{M}(V) \in O\right] \leq e^{\epsilon} \Pr\left[\mathcal{M}(V') \in O\right] + \delta, \tag{6}$$

After a per-sample gradient is computed, DP-SGD clips it to a fixed maximum norm $C$ and Gaussian noise is added to the aggregated parameter gradient with standard deviation $\delta C$. The output of the trained model will satisfy $(\epsilon, \delta)$-DP.

**MemGuard.**  Different from DP-SGD, MemGuard does not change the training process. At a high level, it obfuscates the predictions of the target model by adding noises to its output. It is designed to defend against MIA in particular, while DP-SGD deals with all sorts of privacy risks. Assuming an attack model $\mathcal{A}_{MI}$ has been trained with shadow training (Shokri et al., 2017), and $\mathcal{A}_{MI}(T(x), y)$ outputs a confidence score ranging in $[0, 1]$, where $T(x)$ is the prediction of the target model and $y$ is the label for $x$. A sample is considered a member if the score is larger than 0.5 and a non-member if smaller than 0.5. MemGuard has two phases. In Phase 1, it crafts adversarial noise and adds it to $T(x)$ to force $\mathcal{A}_{MI}(T(x), y)$ to be 0.5 to confuse the attacker, while the distance between the original prediction and the noisy prediction is minimized. In phase II, the adversary adds the noise to the original prediction with a certain probability of trade-off the utility and privacy.

**MixupMMD.**  Li et al. (2021) found a model vulnerability under MIA relates to the difference between the training and testing accuracy, and they proposed MixupMMD to intentionally reduce

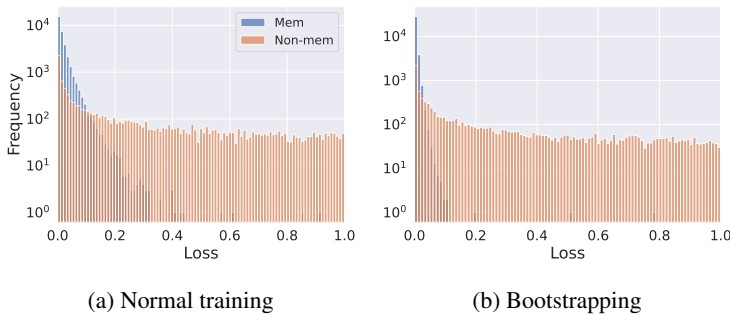

(a) Normal training          (b) Bootstrapping

Figure 8: Loss distribution for models trained on Tiny ImageNet with ResNet-18.

the training accuracy to validation accuracy. A new penalty, Maximum Mean Discrepancy (MMD), is used by the regularizer.

**AdvReg.** Nasr et al. (2018) proposed to mitigate MIA by formulating the defense as a min-max optimization problem. Given a validation set that serves as "non-members", AdvReg introduces an adversarial classifier to infer the membership status using the posteriors generated from the target model. The optimization goal is to minimize the original classification loss and maximize the loss of the adversarial classifier.

## E    ALGORITHMS

We show the CL algorithm in Algorithm 1 and the attack algorithm of our designed attack in Algorithm 2.

---

**Algorithm 1:** Curriculum learning framework.

**Input:** Training dataset $\mathcal{X} = \{\mathbf{X_i}\}_{\mathbf{i=1}}^{\mathbf{N}}$, difficulty measurer $\mathbf{f}(\mathcal{X}, \mathbf{C})$, pacing function $\mathbf{g}(\mathbf{i})$,
      number of iterations $\mathbf{M}$, number of epochs $\mathbf{E}$, target model $\mathbf{T}$

1   $\mathcal{X} \leftarrow f(\mathcal{X}, C)$;
2   **for** $e \in 1, \ldots, E$ **do**
3     **for** $i \in 1, \ldots, M$ **do**
4       $\mathcal{X}'_i \leftarrow \mathcal{X}[1, \ldots, g(i)]$;
5       $\mathcal{B}_i \leftarrow sample(\mathcal{X}'_i)$;
6       $T \leftarrow train(T, \mathcal{B}_i)$

---

**Algorithm 2:** Training the attack model and adjusting threshold under Diff-Cali. "pred" is "prediction".

**Input:** Target model $T$, reference model $S$, shadow dataset $\mathcal{D}$, labels of shadow dataset $L$,
      attack model $\mathcal{A_{MI}}$, curriculum $C$, number of epochs $E$

1   **for** $e \in 1, \ldots, E$ **do**
2     $s_{cal}(T, \mathcal{D}) = s(T, \mathcal{D}) - s(S, \mathcal{D})$;
3     $\theta_0 = \underset{\theta}{arg\,max}\ \text{pred}(\mathcal{A}_{MI}, L, s_{cal}(T, \mathcal{D}))$;
4     $\mathcal{A}_{MI} \leftarrow train(\mathcal{A}_{MI}, s_{cal}(T, \mathcal{D}), g(x, C, \theta_0))$;

---

## F    MORE RESULTS

### F.1    LOSS DISTRIBUTION

The previous evaluation presents a macro-level understanding of CL's impact on MIA. Here we present a micro-level analysis by examining the loss distribution between members and non-members in models trained with normal and CL methods. Due to the space limitation, here we

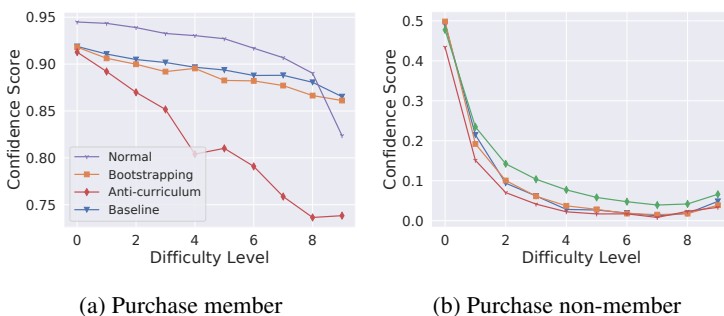

(a) Purchase member        (b) Purchase non-member

Figure 9: Attack model's confidence score for both member and non-member samples on Purchase. MLP is used for target model training, and data samples are arranged according to their difficulty scores from bootstrapping.

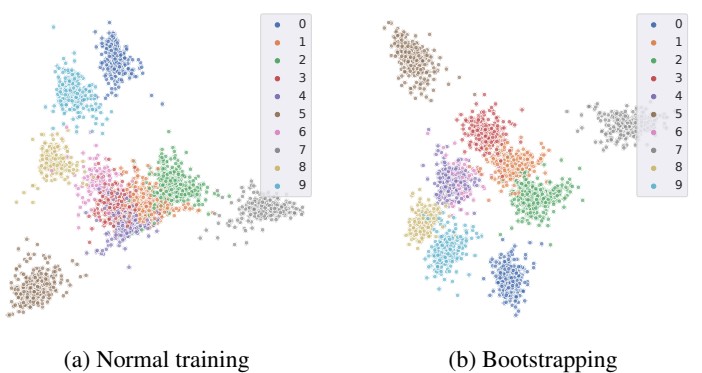

(a) Normal training        (b) Bootstrapping

Figure 10: t-SNE of the classification results on the difficult batch of SVHN.

only show the results of ResNet-18 trained on Tiny ImageNet in Figure 8. There is a clear difference between their loss distributions, e.g., bootstrapping makes the overall members' loss much lower and the members' loss distribution less overlapped with non-members', especially for those members with higher difficulty levels. In Section 5.2, we also reason this observation from the perspective of data memorization.

### F.2 NON-IMAGE DATASETS

As shown in Table 1, most experiments remain to have the same trend they are showing in image datasets. For Purchase, however, attack accuracy on normal training is 0.71% higher than bootstrapping for example. This shows that CL does not always empower MIA more. In Figure 9, we show the confidence score of members and non-members on Purchase, and the result is similar to the image datasets, where difficult samples are more vulnerable.

In the meantime, we found the changes caused by different CL methods are more drastic on the non-image datasets, compared to the image datasets. For example, Texas has a more prominent attack accuracy drop (8.0%) on anti-curriculum. The non-image datasets are relatively simple, containing only binary features after pre-processing, hence they are more likely to be impacted by CL. Table 3 also shows the target model accuracy varies more for the non-image datasets under CL.

### F.3 T-SNE STUDY

To investigate the disparate impact CL has on the classification accuracy across samples. we use t-distributed stochastic neighbor embedding (t-SNE) to visualize the classification tasks carried out by bootstrapping and normal ML on the most difficult batch of data of SVHN. Figure 10 shows all

| Architecture \ Method | Normal | Bootstrapping | Anti-curriculum | Baseline | Transfer Learning |
|---|---|---|---|---|---|
| ResNet-18 | $0.8572 \pm 0.0011$ | $\textbf{0.8751} \pm 0.0001$ | $0.8376 \pm 0.0002$ | $0.8582 \pm 0.0001$ | $0.8718 \pm 0.0001$ |
| ResNet-34 | $0.8564 \pm 0.0001$ | $\textbf{0.8746} \pm 0.0003$ | $0.8481 \pm 0.0002$ | $0.8559 \pm 0.0002$ | $0.8715 \pm 0.0002$ |
| MobileNet | $0.7979 \pm 0.0001$ | $0.8308 \pm 0.0000$ | $0.7763 \pm 0.0002$ | $0.8318 \pm 0.0000$ | $\textbf{0.8430} \pm 0.0001$ |

Table 5: The average accuracy of NN-based attacks on models trained on different network architectures with CIFAR100.

samples within the difficult batch, and it turns out bootstrapping can separate samples from group "1", "2" and "3" better than normal training.

### F.4  TARGET MODEL ARCHITECTURES

To study the impact of the architecture of the target model, we launched MIA against ResNet-34 and MobileNet and compare the results against ResNet-18. Table 5 demonstrates the average attack accuracy of MIA when target models are trained with ResNet-18, ResNet-34, and MobileNet, respectively. It shows that they all share a similar trend of how CL affects MIA. Though MobileNet turns out to be less vulnerable ($5.85\%$ and $5.93\%$ less attack accuracy compared to ResNet-34 and ResNet-18, respectively), bootstrapping, transfer learning, and baseline all increase the overall attack accuracy.

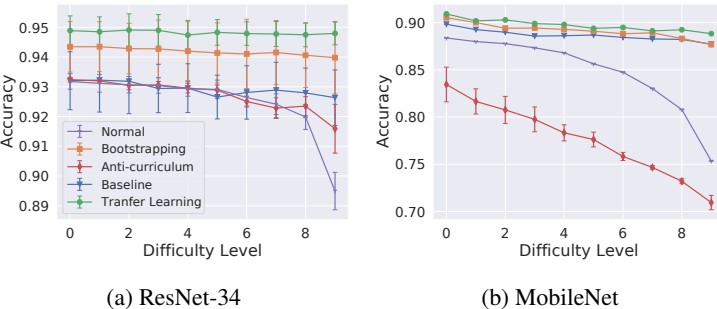

(a) ResNet-34                           (b) MobileNet

Figure 11: MIA accuracy for target model trained on Tiny ImageNet with ResNet-34 and MobileNet, respectively.

Figure 11 demonstrates the results by difficulty levels on ResNet-34 and MobileNet when training with Tiny ImageNet, which can be viewed together with Figure 1b about ResNet-18. Though MobileNet turns out to be less vulnerable ($4\%$ less attack accuracy compared to ResNet-34 and ResNet-18), bootstrapping, transfer learning and baseline all increase the overall attack accuracy, and narrows down the gap between difficult and easy samples. As such, the privacy concerns in CL cannot be addressed by changing target models' architectures. This observation is consistent with other works (Li & Zhang, 2021; He & Zhang, 2021) about MIA vs. architectures.

### F.5  CONFIDENCE SCORE AND ACCURACY FOR MIA AND AIA

**Confidence Score.** Figure 12 and Figure 13 show the confidence scores of the NN-based MIA and Diff-Cali.

**MIA Accuracy.** Figure 14 shows the results on datasets other than the ones used in the main text.

**AIA Accuracy.** Figure 15 shows that CL does not necessarily make the target model more vulnerable to AIA.

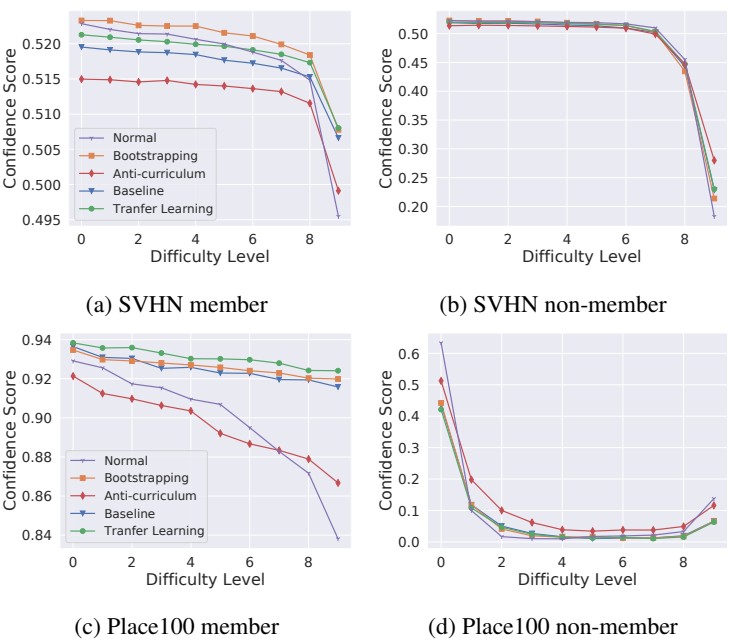

Figure 12: Attack model's confidence score for both member and non-member samples. ResNet-18 is used for target model training, and data samples are arranged according to their difficulty scores from bootstrapping.

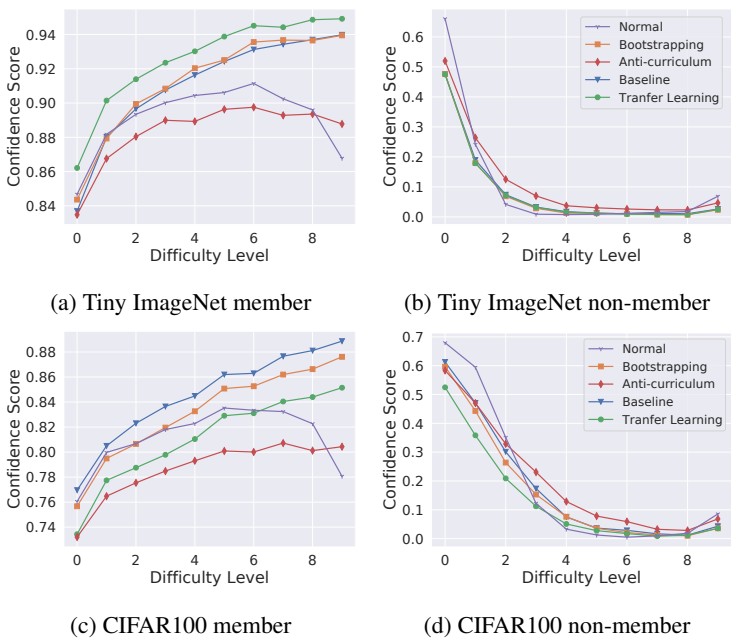

Figure 13: Diff-Cali's member and non-member confidence score for ResNet-18.

## F.6 EVALUATION OF DEFENSE

We evaluate how the defenses, including DP-SGD, MemGuard, MixupMMD, and AdvReg perform under the impact of CL. Table 6 shows the attack accuracy on ResNet-18 which is trained with CIFAR100. Because MixupMMD and AdvReg require reference datasets for defense deployment, we equally divided CIFAR100 into 5 parts for fair comparison among all the defense techniques.

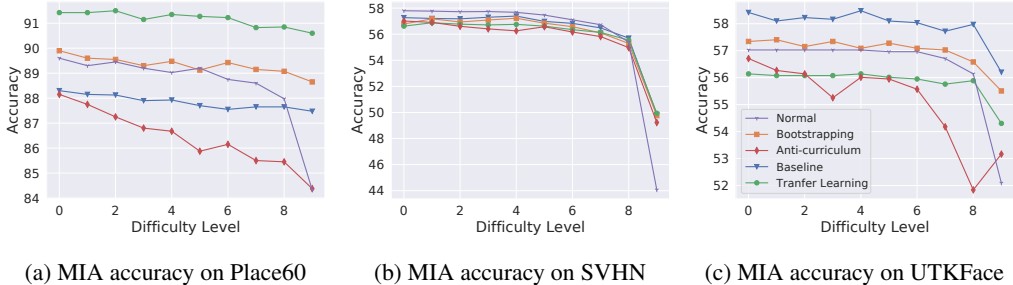

(a) MIA accuracy on Place60  (b) MIA accuracy on SVHN  (c) MIA accuracy on UTKFace

Figure 14: MIA accuracy on Place60, SVHN and UTKFace. ResNet-18 is used for target model training and bootstrapping is used for CL. The x-axis represents the difficulty level of the data, and the y-axis represents the attack accuracy.

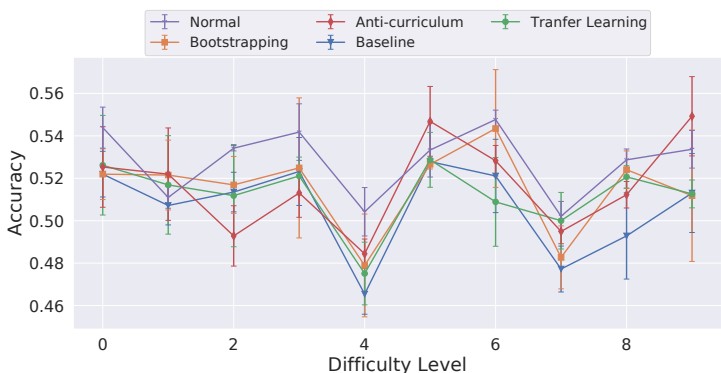

Figure 15: Attribute inference attack accuracy on UTKFace

More specifically, all target models in Table 6 are trained with only $12,000$ data points, which also explains why the accuracies are lower.

Regarding the setup of the defense methods, bootstrapping and anti-curriculum with DP-SGD are trained with the same curriculum as previous experiments. DP-SGD* uses a *noisy* curriculum for bootstrapping and anti-curriculum, and the difficulty measurer is trained under DP-SGD. For transfer learning, it is not impacted as we use a pre-trained model. $\epsilon$ and $\delta$ in our evaluation are $124,496$ and $1e-5$ for DP-SGD. We have a large $\epsilon$ because we have 200 epochs of training and ResNet-18 contains a large number of parameters. We did not change these settings for a fair comparison with other defense techniques. Previous studies have used large $\epsilon$ for DP-SGD in order to achieve good model accuracy (Jayaraman & Evans, 2019; Kurakin et al., 2022). Based on a recent work Bu et al. (2022), we are able to make $\epsilon$ 10 times smaller after proper parameter tuning while achieving similar target accuracy. The $\epsilon$ can be brought down even first with large batch size. Pulling tricks of DP-SGD based on the above recent work can further boost the tradeoff, we do not discuss it here as that is a parallel line of research. Note that in this section, we still use small batch size for DP-SGD evaluation though that results in large $\epsilon$. This is because we want to keep parameters across all target models the same for a fair MIA evaluation, and we have limited computing resources for handling large batch number.

Table 6 demonstrates that DP-SGD is able to curb the MIA accuracy from 90.8% to 50.5% in average, which is close to random guess (i.e., member or non-member), though at the cost of a significant drop in target model's classification accuracy (from 49.52% to 16.42% in average). This observation is consistent with previous works (Li et al., 2021; Kurakin et al., 2022). We also found DP-SGD is effective against Diff-Cali (e.g., attack accuracy for normal and bootstrapping are dropped to 53.67% and 53.09%). For DP-SGD*, due to the introduced noise, the ranking given by its curriculum is less accurate, but Table 6 shows that such change does not impact the MIA accuracy, and the target model accuracy drops by only a small amount (i.e., 0.8% for bootstrapping and 0.7% for baseline)

|  | None | | DP-SGD* | | DP-SGD | | MemGuard | | | MixupMMD | | AdvReg | |
|---|---|---|---|---|---|---|---|---|---|---|---|---|---|
|  | Target | MIA | Target | MIA | Target | MIA | Target | MIA | Label-only | Target | MIA | Target | MIA |
| norm | 48.0 | 90.3 | 17.4 | 50.6˙±0.11 | 17.4 | 50.8˙±0.07 | 48.0 | 50.0 | 83.0 | 54.1 | 81.6˙±0.02 | 51.2 | 89.2˙±0.01 |
| bstp | 51.4 | 91.4˙±0.03 | 18.0 | 50.6˙±0.06 | 17.2 | 50.7˙±0.01 | 51.4 | 50.0 | 84.5 | 54.4 | 83.1˙±0.02 | 54.2 | 91.6˙±0.02 |
| tran | 48.9 | 91.3˙±0.03 | 17.2 | 50.6˙±0.01 | 17.2 | 50.6˙±0.01 | 48.9 | 50.0 | 84.5 | 55.7 | 76.1˙±0.03 | 50.4 | 92.8˙±0.04 |
| base | 50.0 | 91.5˙±0.02 | 18.3 | 50.4˙±0.11 | 17.6 | 50.4˙±0.11 | 50.0 | 50.0 | 84.0 | 55.0 | 84.4˙±0.02 | 53.0 | 91.6˙±0.01 |
| anti | 49.3 | 89.5˙±0.02 | 11.2 | 50.3˙±0.11 | 17.2 | 50.4˙±0.10 | 49.3 | 50.0 | 81.3 | 52.6 | 79.1˙±0.02 | 52.1 | 87.3 |

Table 6: The average accuracy of MIA ($\pm$ standard deviation (STD)) on target model trained on CIFAR100 with ResNet-18 and different defense methods. All numbers are in percentage, entry without $\pm$ STD means the STD is less than 0.01%.

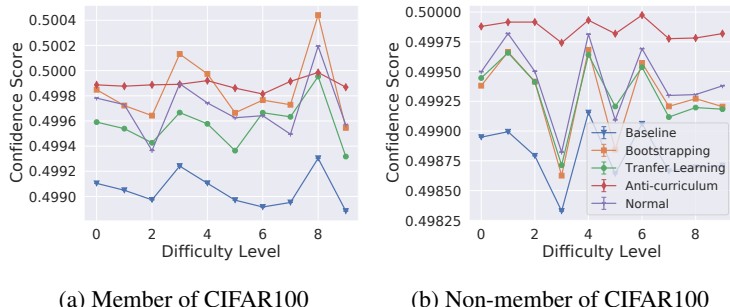

(a) Member of CIFAR100      (b) Non-member of CIFAR100

Figure 16: Attack model's confidence score for member and non-member samples of CIFAR-100 trained on ResNet-18 with DP-SGD.

except for anti-curriculum. Due to the noise in ranking, the ranking for anti-curriculum is no longer strictly ordered from difficult to easy. Instead, it becomes more random, thus target accuracy of anti-curriculum is even closer to baseline or bootstrapping. In general, the result suggests using noisy ranking (DP-SGD*) as a defense might not be effective.

For MemGuard, due to its design, NN-based MIA accuracy is fixed to 50% when the defender knows what MIA method is performed by the attacker. In the meantime, the classification task of the target model is not impacted by MemGuard. However, it is not very effective towards label-only attack, as it does not change the label. Our evaluation shows that the overall label-only attack accuracy can still reach up tp 86% even with MemGuard deployed. MixupMMD decreases the MIA accuracy (e.g., 91.4% to 83.1% for bootstrapping), and interestingly, it increases the target model accuracy (e.g., from 51.4% to 54.4% for bootstrapping), which might be attributed to its new regularizer. AdvReg can also increase target accuracy (e.g., 51.4% to 54.2% for bootstrapping) but is less effective in mitigating MIA (e.g., MIA accuracy is even increased from 91.4% to 91.6% for bootstrapping). This observation concurs with a previous work Song & Mittal (2021).

Given that CL introduces disparate impact on samples under different difficulty groups, we further investigate the relation between difficulty groups and defenses, and we focus on DP-SGD. Figure 16 shows that DP-SGD is able to eliminate the disparate impact by CL, essentially making the difficult samples again hard to attack. We speculate the reason is that DP-SGD adds noise to gradient, which adds randomness to the optimization phase. CL, by introducing a teacher module, reinforces the learning by reducing the randomness. Ultimately, DP-SGD and CL are built on two opposite foundations. Thus, DP-SGD can eliminate the benefit from CL and achieve significant defense effect.

Overall, there is still room for improvement in defenses. Potential future work is to preserve certain properties brought by an ML technique (e.g., fast convergence and higher final performance by CL) and mitigate privacy risks generically.

