# OpenReview forum: "A Comprehensive Study of Privacy Risks in Curriculum Learning"
_ICLR.cc/2024/Conference — Submitted to ICLR 2024_

### Official Review · Reviewer_rVvZ · 2023-10-28

**Soundness:** 2 fair
**Presentation:** 2 fair
**Contribution:** 2 fair
**Rating:** 3
**Confidence:** 4

**Summary:**

This paper studies the privacy risks in curriculum learning (CL). The authors perform membership inference attacks and attribute inference attacks against four curricula designs and normal training on 9 datasets. They draw several conclusions on the privacy risks induced by CL as well as the disproportional vulnerability of different samples. A new MIA is proposed based on calibrating the difficulty scores.

**Strengths:**

- Comprehensive and thorough evaluation of privacy risks in a concrete learning setting
- The proposed MIA seems promising

**Weaknesses:**

My general complaint is that this paper feels too much of a technical report rather than a research paper that I would expect to see at a top ML conference. I will elaborate:

- **The motivation is not well-established**. The authors mentioned "it’s crucial to investigate how training techniques affect privacy" -- I generally agree with this point, but this is not sufficient to convince me that studying the privacy risk in curriculum learning is important. 1) From a practical perspective, curriculum learning is a relatively niche subject, and is very different from self-supervised learning or unsupervised learning which have attracted extensive interest in the ML community. I'm not aware of any of its real-world applications and am not sure whether privacy is indeed a valid issue within. 2) From a research perspective, the author briefly mentioned "data ordering could have negative impacts on privacy" -- this is a valid hypothesis, and could be a good starting point to perform some thought experiments to obtain a more fine-grained argument (e.g., what kind of data ordering, what is the connection to curriculum learning, what data points will be affected the most). Directly jumping to large-scale evaluations based on an unmature hypothesis seems too hasty to me.

- **The structure is weird**. 1) Sec 3 (dataset and model) should be placed right before Sec 5 (the evaluations results), and I feel a paragraph (rather than a section) should be sufficient; 2) the proposed MIA (Diff-Cali) is mixed with other background knowledge on MIAs and AIAs in Sec 4, which is super strange; 3) there are no defense results in the main text yet the defense strategies are mentioned in the abstract, intro and conclusion. Overall this leaves me a strong feeling that the paper contains too many contents and is not well-compiled for reading.

- **Many technical details are missing**. The authors put too many technical details to the appendix, for instance, the MIAs, the defense strategies, and TPR at low FPR. Even the details of curriculum learning is not clearly explained. This will significantly hinder understanding even for people working on the intersection of ML and privacy. Combined with the previous point, I think this paper is more suited for a security conference.

- **The analyses are not sufficiently in-depth**.  Most of the conclusions can be directly drawn from the experimental results, and there are few insights or discussions beyond. For instance, I did not learn much from the AIA experiments beyond knowing that the attack success rate does not increase, and the hypothesis "sensitive attribute to be inferred is not influenced by data ordering and repeating" is not backed up by any further evidence. In a few places where the authors tried to offer more principled analysis, I observed a few reasoning gaps. For instance, the authors attempted to use memorization to explain the impact of curricula. However, there is a significant gap between the experimental setting in Sec 5.2 and the actual curriculum learning. More specifically, for curriculum learning as described in algorithm 1, hard samples appear later than easy samples *only in a single epoch*; they do appear in every epochs (and particularly, early epochs) of training.

**Questions:**

- In algorithm 1, it seems that for later iterations in a given epoch (i.e., large $i$), $X_i'$ will still contain easy samples. This suggest that easy samples will still appear in later iterations of training (since $B_i$ is sampled from $X_i'$), meaning that they will have high occurrence compared to hard samples. 1) I'm not familiar with curriculum learning, but I feel a more natural way is to segment the ordered samples (based on difficulty scores) into *disjoint* batches, and then progressively train on them in a single epoch; 2) If you indeed used algorithm 1, then the number of occurrence will be another factor (other than difficulty) that will impact the vulnerability of each sample, and this is not taken into account when designing experiments or drawing conclusions.

- In Fig 1a and 2a, the harder samples actually have lower attack accuracy and confidence scores (for all curricula and normal training). This is very counter-intuitive and not explained. Particularly, I don't buy the results that harder samples will have low attack success rate for normal training. This makes me doubt that the difficult scores used in this paper is problematic. Also, I think it is misleading to claim that "harder samples are more vulnerable". Instead, what you can actually infer from the results is that "CL will introduce stronger effects to harder samples compared to normal training" (i.e., the *gap* is large).

---

> ### Author Response · Authors · 2023-11-17
> **Thanks for writing such detailed comments for us. We appreciate your time!**
>
> **Motivation**: First, We argue that this work is well-motivated, as evidenced by the application of CL in emerging language models [1]. Researchers have proposed a CL approach termed 'length-based CL,' which uses the length of input text as a measure of difficulty.
>
> Secondly, the study referenced in [2] investigated data ordering for data poisoning in machine learning, confirming that this concept is more than just a hypoyhesis. While “data ordering” might seem vague, its definition in the context of CL is well-defined— it refers to the sequence in which data are presented to the model. This definition aligns with that in [2], lending further support to our motivation for investigating the potential negative impacts of such specific data ordering. Regarding your comment about 'which data points will be affected the most’, identifying these is one of the goals of our study. We began with the overarching question of how CL influences model privacy in general and then proceeded to a sample-wise investigation to determine which types of data points are most susceptible.
>
> **Structure and Technical details**: See official comment on top please.
>
> **Analysis**: We argue that our results regarding AIA, as shown in Figure 6 (which will be updated to Table 3) and Figure 15, demonstrate both overall and sample-wise privacy investigations on AIA. The fact that AIA does not indicate increased vulnerability in CL simply suggests that CL does not fundamentally alter how the target model captures irrelevant (or, in our specific example, sensitive) attributes.
>
>
> **Question 1**: Both definitionsare recognized in curriculum learning literature[3]. We specifically use 2) and have previously tested out 1) too, our preliminary results showed that they lead to same conclusion.
>
> I find the concept of sample occurrence very interesting and would love to add more discussion and experiment for it. However, we argue that occurrence is closely linked to difficulty; generally, a higher occurrence implies easier samples (based on your listed definition 2) of CL). Based on our current results, data ordering exerts a more significant influence on performance than the frequency of data occurrence. For example, in normal training (as shown in Figure 1), hard samples appear in all epochs, while in Curriculum Learning (bootstrapping and transfer learning), there hard samples appear less especially in the early epochs. Nonetheless, the MIA accuracy for hard samples in CL (e.g., samples with difficulty level of nine) still surpasses that in normal training.
>
> To further explore the impact of occurrence, especially on easier samples which present a less intuitive case, we plan to train a CL model with disjoint batches. This approach will help us isolate and assess the effect of occurrence. It's important to note that our preliminary results indicate that using disjoint batches leads to the same key conclusions as presented in this paper. This implies that the occurrence is not the primary factor causing vulnerability in CL; rather, it is the data ordering. Samilar conclusion can be drawn from Figure 1 for the hard samples (see last paragraph).
>
>
> **Question2**: We're interested in understanding why you find the concept of 'harder samples having a lower attack success rate in normal training' counter-intuitive. Could you please elaborate on that?  To explain, let's first define what makes a sample 'difficult.' A common measure of difficulty is the loss score of a sample, where a higher loss score indicates greater difficulty. In the context of transfer learning, this score is derived from a pre-trained model. A high loss score could mean that the pre-trained model struggles to 'remember' this sample, possibly because the sample is rare, its features are too distinctive, or for other reasons. If a target model has difficulty remembering a sample, it's logical to assume that such a sample would be less vulnerable to attacks, as the model reveals less information about it.
>
> We also want to clarify that when we say 'harder samples are more vulnerable,' we specifically refer to their vulnerability in the context of CL compared to a non-CL setting. This might be more an issue of how we present our ideas. Phrasing it as 'CL introduces stronger effects on harder samples compared to normal training' seems to convey the same idea in my opinion. However, we're open to rephrasing this for greater clarity for the audience.
>
> [1] Length-Based Curriculum Learning for Efficient Pre-training of Language Models. New Gen. Comput. 41, 1 (Mar 2023)
>
> [2] Manipulating sgd with data ordering attacks. Advances in Neural Information Processing Systems.
>
> [3] Petru Soviany, Radu Tudor Ionescu, Paolo Rota, Nicu Sebe: Curriculum Learning: A Survey.

---

### Official Review · Reviewer_zNpb · 2023-11-01

**Soundness:** 3 good
**Presentation:** 3 good
**Contribution:** 2 fair
**Rating:** 5
**Confidence:** 4

**Summary:**

This paper empirically study the privacy risks introduced by CL, by launching a series of privacy attacks in the forms of MIA and AIA, and privacy defenses. The authors also invented a new MIA method.

**Strengths:**

The paper is easy-to-read and comprehensive from an empirical point of view. It tackles an important topic with some convincing results. I specifically like the discussion of model forgetting below Figure 2 (more discussion on this please).

**Weaknesses:**

This paper leverages existing CL, MIA/AIA methods and defenses, thus falling short on the novelty. This choice of tools also make the results less exciting. While some novelty is indeed introduced by the new MIA method, its performance is similar to NN-based MIA (I quote "...Diff-Cali achieves slightly lower (less than 1.44%) accuracy compared to NN-based attack"), i.e. the difference is not significant even if the difficult samples are made more vulnerable.

Minor: Table 1 & 2 inconsistent values on CIFAR100.
Figure 6 is not a Figure at all.

**Questions:**

See Weaknesses.

---

> ### Author Response · Authors · 2023-11-17
> **We appreciate your time reviewing our paper.**
>
> **Weakness and Question**:
> We argue that even though our choice of tools and Curriculum Learning (CL) approaches are based on existing works, we are the **first** to study the privacy risks introduced by CL. Furthermore, we believe the prominence of privacy concerns is more convincingly demonstrated if existing MIA can already cause privacy issues in CL, not to mention the potential impact of newer attack methods.
> We found that Diff-Cali is extremely effective in terms of True Positive Rate (TPR) at low False Positive Rate (FPR), achieving over 50% for all samples compared to the NN-based attack. This effectiveness is significant, considering the importance of TPR at low FPR as a metric. As [1] states, “This focus (on the low-false positive rate regime) is the setting with the most practical consequences. For example, to extract training data, it is far more important for attacks to have a low false positive rate than a high average success rate, as false positives are far more costly than false negatives. Similarly, de-identifying even a few users contained in a sensitive dataset is far more significant than making an average-case statement like ‘most people are probably not contained in the sensitive dataset’”.
>
> **Minor**: The entry in Table 2 contains a typographical error and should read 0.8577; additionally, Figure 6 has been changed to Table 3.
>
>
> [1] “Membership Inference Attacks From First Principles” in IEEE S&P 2022

---

### Official Review · Reviewer_tgat · 2023-11-01

**Soundness:** 3 good
**Presentation:** 3 good
**Contribution:** 2 fair
**Rating:** 6
**Confidence:** 4

**Summary:**

This paper explores the privacy risks associated with Curriculum Learning (CL). The paper examines Membership Inference Attacks (MIAs) and Attribute Inference Attacks (AIAs). Under the CL setting, the paper reveals a slight increase in vulnerability to MIA, a vulnerability not observed with AIA. This paper also proposes a new MIA method, termed Diff-Cali, by investigating the attack performance of MIA on varying samples. The paper also evaluates several defense mechanisms for their efficacy in mitigating these risks.

**Strengths:**

1. This paper is the first work to study the privacy risks introduced by CL and does a comprehensive comparison of different settings including different attack types, target models, and datasets.
2. This paper also studies the performance of different defense methods to reduce the success rate of privacy attacks under curriculum learning.

**Weaknesses:**

1. This paper limits the datasets to image and tabular datasets. It will be interesting to study other data types like text and graphs, to make this study more comprehensive.
2. While the paper evaluates various defense methods, it does not have a deeper investigation into why certain defenses underperform and how they could be potentially improved.

**Questions:**

1. Could the proposed Diff-Cali method be combined with existing defense mechanisms to create a more robust defense against privacy attacks in Curriculum Learning?
2. How would the introduction of federated learning or decentralized training scenarios affect the privacy risks and defense mechanisms evaluated in this study?

---

> ### Author Response · Authors · 2023-11-17
> **Thank you for your review! We appreciate it!**
>
> **Weakness 1**: We can possibly add text classification task upon request with data set from torchtext.datasets to this study to make it more comprehensive. More specifically, we plan to use the example shown in [1] for this study.
>
> **Weakness 2**: In general, many of these defenses operate under different assumptions and vary in effectiveness depending on the setting.
>
> DP-SGD trains models with noise, which fundamentally reduces the model’s ability to memorize individual samples, thus leading to superior defensive performance.
>
> MemGuard, on the other hand, alters a model’s output to defend against MIA. This is the reason that it doesn’t affect the target model yet can still be effective. However,  it also does not offer protection against label-only attacks due to its lack of changes to the target model.
>
> MixupMMD and AdvReg both modify the posterior to enhance privacy. MixupMMD is more effective because it brings the member and non-member posterior distributions even closer than AdvReg by introducing the new penalty  termed Maximum Mean Discrepancy.
>
> From all the defenses mentioned above, we've learned that a key factor in making a defense effective is to minimize the difference between member and non-member posterior distributions. Therefore, a potential direction for future research in improving defenses could involve exploring other methods to further align these posterior distributions. We plan do add this discussion to the paper.
>
> **Question 1**: Could you please further elaborate on this question? If we understand your question correctly, are you asking whether modifying the difficulty score can make Curriculum Learning (CL) less vulnerable to attacks, and whether the difficulty score can be utilized as part of a defense strategy? Based on our study across various CL settings (e.g., baseline), we found that even with randomized difficulty scores, CL remains more vulnerable compared to normal training. Therefore, the key to a more robust defense lies in minimizing the gap between member and non-member posterior distributions.
>
> **Question 2**:
> We consider federated learning and decentralized training a parallel line of study to ours. However, when combined with our study, the key conclusions remain the same for all decentralized models, and we believe the central model will be less intrusive compared to our findings. This is because, fundamentally, federated learning and decentralized training are designed to offer enhanced privacy.
>
>
> [1] https://github.com/pytorch/text/tree/main/examples/text_classification

---

### Official Review · Reviewer_455n · 2023-11-03

**Soundness:** 2 fair
**Presentation:** 3 good
**Contribution:** 3 good
**Rating:** 5
**Confidence:** 4

**Summary:**

The paper studies the privacy leakage of CL through a comprehensive evaluation of CL using different membership inference attack and attribute inference attacks using nine benchmarks.

**Strengths:**

1. The paper are well written and the problem is well-motivated.
2. The experiment are comprehensive and some empirical findings are insightful (e.g., CL cause disparate impact to members and non-members is useful)

**Weaknesses:**

1. The paper misses an important SOTA NN-based MIA attack LiRA (from the paper “Membership Inference Attacks From First Principles” in IEEE S&P 2022), making the evaluation of NN-based less convincing.
2. Difficulty Calibrated MIA is an incremental improvement over Watson et al. 2022 via adaptatively changing $\theta$ based on the difficulty level.
3. The author does not justify the threat model clearly. For example, the Difficulty Calibrated MIA implicitly assumes the target model training is using CL. However, such type of information is not usually available to the attack. In this case, the author might also want to evaluate the proposed attack under a non-CL setting and compare its performance with other MIA attacks.

**Questions:**

1. In the defense part, it is natural to extend the idea of allocating privacy budget accordingly based on the difficulty level of the training examples. Have you tried this in the defense evaluation?

---

> ### Author Response · Authors · 2023-11-17
> **Thank you for your valuable comments.**
>
> **W1-LiRA**: In general, not all attack methods (e.g., adaptive attack and LiRA) and defense techniques have been examined in our paper. Although LiRA is considered state-of-the-art, it requires lots of shadow models (for example, 256 shadow models for CIFAR100, as demonstrated in [1]), whereas all other attacks in our paper require only one. To compare fairly with LiRA, we would need to divide the current datasets into much smaller subsets, which would likely degrade the performance of all target and shadow models. Moreover, training 256 shadow models for both CL and no-CL settings would be very costly. Due to these reasons, LiRA was not investigated in this study. However, we believe our key findings (e.g., the increased vulnerability of difficult samples when trained with CL) are broadly applicable, given the fundamental principles of curriculum design.
>
> Furthermore, we can possibly run a downscaled version of LiRA upon request (i.e., with fewer shadow models) in a case study to demonstrate that our key conclusions remain valid. More specifically, we intend to re-divide the dataset into six subsets, $D_i$  where $i = 0,1,2,3,4,5$. We will use $D_0$ to train the target model, $D_1$, $D_2$, $D_3$, and $D_4$ to train shadow models for LiRA and NN-based methods, and then use $D_5$ as the target and shadow test dataset.
>
> **W2**: From the technical perspective, Diff-Cali only made a small change. However, its performance proves to be extremely effective in bringing the True Positive Rate (TPR) at a low False Positive Rate (FPR) above 50%. Thus, we argue that it is effective, even though we didn’t make any major changes. Furthermore, we argue that TPR at low FPR is an extremely important metric. Quoting from [1], “This focus (on the low-false positive rate regime) is the setting with the most practical consequences. For example, to extract training data, it is far more important for attacks to have a low false positive rate than a high average success rate, as false positives are far more costly than false negatives. Similarly, de-identifying even a few users contained in a sensitive dataset is far more significant than making an average-case statement like ‘most people are probably not contained in the sensitive dataset’”.
>
>
> **W3**: We tested Diff-Cali in both non-CL settings and various CL settings (refer to Figures 3 and 5). In these figures, we used bootstrapping as the default curriculum referenced in Diff-Cali and then applied this attack to models trained with all non-CL and CL methods (the "normal" lines in Figures 3 and 5 represent the non-CL setting). As observed, difficult samples become more vulnerable in all cases, regardless of whether CL is used in the target model. However, when the matching CL is applied, the improvement in accuracy on difficult samples is more pronounced. For instance, comparing Figure 1a with Figure 5a, the improvement in accuracy at difficulty level 9 is approximately 2.5 for anti-CL, while for bootstrapping, it is nearly 7.5.
>
> **Question-Defense**: We have not yet explored this, but we believe the fundamental conclusions would remain unchanged, given DP-SGD's proven success in defending against MIA. Allocating the privacy budget based on the difficulty level could potentially yield a better balance between privacy and utility. However, this approach would impact the moments accountant technique used by DP-SGD for total budget accounting and require an examination of privacy composition. Additionally, incorporating the difficulty level might alter the privacy guarantee, possibly requiring the addition of noise to this difficulty level as well.
>
>
> [1] “Membership Inference Attacks From First Principles” in IEEE S&P 2022

---

> > ### Comment · Reviewer_455n · 2023-11-23
> > **Response after Rebuttal**
> >
> > After reading the authors' response and other reviewers' comments, my standing towards the paper remains the same. The technical depth of the paper, the motivation of the paper, and technical details + analysis makes the paper a borderline paper. Thus, I will keep my score.

---

### Author Response · Authors · 2023-11-17
**To address paper structure and incorporate more technical details, we'd appreciate everyone's input!**

We acknowledge the feedback from  Reviewer rVvZ regarding the potentially confusing structure of our paper, stemming from our attempt to cover a broad range of content. **We understand that the audience’s interests can be highly subjective, and in response, we welcome specific guidance from the reviewers. We would appreciate knowing the top three aspects or findings about our work that you find most compelling.** Based on your input, we are prepared to restructure our paper to highlight these key areas and reintegrate more technical details, aligning with what the reviewers deem most critical or interesting. We believe that ICLR is a suited venue for our work, especially given the increasing focus on LLMs and the application of CL in emerging language models [1].

In summary, we plan to refine our paper by combining the reviewers' advice to present the most intriguing findings to the audience. An extended version of the paper will also be made available for those interested.

---

### Meta-Review · Area_Chair_8NgL · 2023-12-05

**Metareview:**

Unfortunately, the reviewers were not excited about the paper. In particular, there were concerns regarding the motivation for studying  privacy risks in curriculum learning. Also there were concerns about the baselines considered in the paper, which the reviewers believed to be present even after the rebuttal.

**Justification For Why Not Higher Score:**

The reviewers were not excited about the paper.

**Justification For Why Not Lower Score:**

NA

---

### Decision · Program_Chairs · 2024-01-16

Reject